# Assessment of Landslide Susceptibility Combining Deep Learning with Semi-Supervised Learning in Jiaohe County, Jilin Province, China

**Jingyu Yao, Shengwu Qin \*, Shuangshuang Qiao, Wenchao Che, Yang Chen, Gang Su and Qiang Miao**

College of Construction Engineering, Jilin University, Changchun 130026, China; yaojy18@mails.jlu.edu.cn (J.Y.); qiaoss17@mails.jlu.edu.cn (S.Q.); chewc18@mails.jlu.edu.cn (W.C.); yangchen18@mails.jlu.edu.cn (Y.C.); sugang18@mails.jlu.edu.cn (G.S.); miaoqiang18@mails.jlu.edu.cn (Q.M.)
* Correspondence: qinsw@jlu.edu.cn

**Abstract:** Accurate and timely landslide susceptibility mapping (LSM) is essential to effectively reduce the risk of landslide. In recent years, deep learning has been successfully applied to landslide susceptibility assessment due to the strong ability of fitting. However, in actual applications, the number of labeled samples is usually not sufficient for the training component. In this paper, a deep neural network model based on semi-supervised learning (SSL-DNN) for landslide susceptibility is proposed, which makes full use of a large number of spatial information (unlabeled data) with limited labeled data in the region to train the mode. Taking Jiaohe County in Jilin Province, China as an example, the landslide inventory from 2000 to 2017 was collected and 12 metrological, geographical, and human explanatory factors were compiled. Meanwhile, supervised models such as deep neural network (DNN), support vector machine (SVM), and logistic regression (LR) were implemented for comparison. Then, the landslide susceptibility was plotted and a series of evaluation tools such as class accuracy, predictive rate curves (AUC), and information gain ratio (IGR) were calculated to compare the prediction of models and factors. Experimental results indicate that the proposed SSL-DNN model (AUC = 0.898) outperformed all the comparison models. Therefore, semi-supervised deep learning could be considered as a potential approach for LSM.

**Keywords:** landslide susceptibility; deep learning; semi-supervised learning; DNN; SVM; LR; AUC; IGR

## 1. Introduction

Landslide, as one of the most destructive natural disasters, is caused by the combination of natural and human factors [1–4]. The essence of landslide is described as the movement process of earthen material sliding off the slope due to gravity, which seriously threatens the safety of life and property [5,6]. This type of disaster not only gives rise to the direct damage of living facilities, but also leads to the depletion of land resources [7,8]. In recent decades, a large amount of sloping land has been reclaimed in northeast China, resulting in frequent mountain disasters. Therefore, managing and evaluating the future location of landslides are particularly important in response to such threats. At present, LSM is an attractive way to evade the risk of this disaster [9,10] as the factors that triggered landslides in the past may continue to induce landslides in the future.

The performance of LSM depends on the fitting effect of the model and the quality of the input data [11,12]. The refinement of algorithms has provided an effective tool for the research of susceptibility, and machine learning models have been widely devoted to constructing functional mappings between variables and landslide susceptibility [13,14] including logistic regression (LR) [15], radial basis function (RBF) [16], artificial neural network (ANN) [17], random forest (RF) [18], decision tree (DT) [19], regression tree (RT) [20], and support vector machines (SVM) [21]. Compared with the subjective and heuristic models, the machine learning models can successfully handle non-linear data with different scales in the fields of remote sensing, disaster mitigation and wildfire protection [22,23]. However, these models can be considered as shallow learning structures with only one or zero hidden layers. A large number of shortcomings exist in these models such as limited training time, unstable convergence, local optimal, and so on [24]. In recent years, deep learning (DL), as an attractive framework, has motivated a trend of unprecedented advancement in susceptibility assessment [25]. DL has significant advantages over traditional models: The ability to build advanced features encourages the discovery of the deepest connection between the parameters, which generally obtain a robust performance for nonlinear processing [26,27]. The latest research has revealed that DNN processed high learning potential in landslide susceptibility assessment with different sampling strategies [28]. The DNN models with multiple optimization algorithms such as stochastic gradient descent (SGD), root mean square propagation (RMSProp), and adaptive moment optimization (Adam) have been compared with traditional machine learning models [29], and their excellent performance and applicability have been confirmed for landslide susceptibility.

Nevertheless, in geospatial analysis, the amounts of corresponding labeled samples (landslide or non-landslide events) are still limited and are difficult to collect compared with the huge study area. Particularly for deep learning, the parameters need to be supported by a large number of labeled samples [30]. It is common that the number of disasters usually fails to meet the requirements of modeling. This problem may lead to deviation and over-fitting, which may bring about inestimable errors in the prediction [31,32]. Although the regularization technique and reducing the dimension of feature can screen valuable information to some extent, the expansion of sample data is still the most effective means to enhance training [33,34]. With the development of remote sensing (RS) and geographic information system (GIS) technologies, high-resolution digital elevation models (DEM) and engineering data are more easily obtained. A large number of unlabeled sites are not utilized, which are rich in location and geographic information [35]. Meanwhile, unsupervised learning using only unlabeled data can also implement LSM due to their advantages of strong efficiency and scalability for training [36]. Therefore, how to make full use of the unlabeled information is a feasible direction for the research of LSM.

Semi-supervised learning, a paradigm of machine learning, considers both labeled data and unlabeled data. The core is the assumption that unlabeled samples can provide effective spatial distribution information of features (clustering center estimation) [37]. At present, a great deal of research has been devoted to developing a reasonable framework for training unlabeled data [38,39]. A weakly labeled support vector machine was proposed to assess urban flood susceptibility [35]. Pseudo-label learning can enhance the training process of models using labeled data and pseudo-labels, and the method has been successfully applied to a variety of classification tasks [30]. Automatic encoder and clustering algorithm were applied to realize pseudo-label classification and pre-training of the network [40]. The efficient and high-quality classifiers were built by simple self-training or cooperative training [32,41]. As an end-to-end model of semi-supervised learning method, ladder networks can minimize the loss of supervised and unsupervised learning during the training process [42]. The classification accuracy of unlabeled data determines the effect of semi-supervised learning to a certain extent [43]. The cluster-then-label technique for semi-supervised learning [44], as a typical generation model, uses the unsupervised clustering algorithm to identify the clustering of unlabeled data. More recently, a large number of collaborative learning frameworks combined with deep learning and clustering algorithm [45,46] were constructed to obtain high quality samples and could learn

the parameters and depth characteristics of the network iteratively. The results have proven that semi-supervised learning avoids the waste of data and resources, and that problems can be solved such as the weak generalization ability of supervised learning and the imprecision of unsupervised learning [47]. Therefore, in this study, a semi-supervised learning framework based on deep neural network (SSL-DNN) was proposed for LSM, which attempts to pre-train a network using unlabeled samples based on a K-means clustering algorithm, and then conduct supervised fine-tuning using labeled data. We speculate that the SSL-DNN model can achieve more accurate and stable LSM than the traditional supervised learning models.

In order to verify the assumptions of the article, a landslide susceptibility evaluation was carried out in Jiaohe County in Jilin Province, China. Meanwhile, SVM and LR models were devoted for comparison. The purpose was that a high-quality LSM can be produced to provide a decision basis for early warning of landslide hazards. In addition, the combination of deep learning and semi-supervised learning was first applied in the field of LSM.

## 2. Study Area

Jiaohe County is located in the east of Jilin Province and lies between 43.19° N–44.4° N latitude and 126.75° E—128.01° E longitude. The area is 6429 km$^2$, with a length of 98 km from north to south and 103 km from east to west (Figure 1). The area belongs to the continental monsoon climate in the northern cold temperate zone, with an average annual temperature of 3.6 °C. The perennial average rainfall is 500 mm–700 mm, and the precipitation is concentrated in June to August.

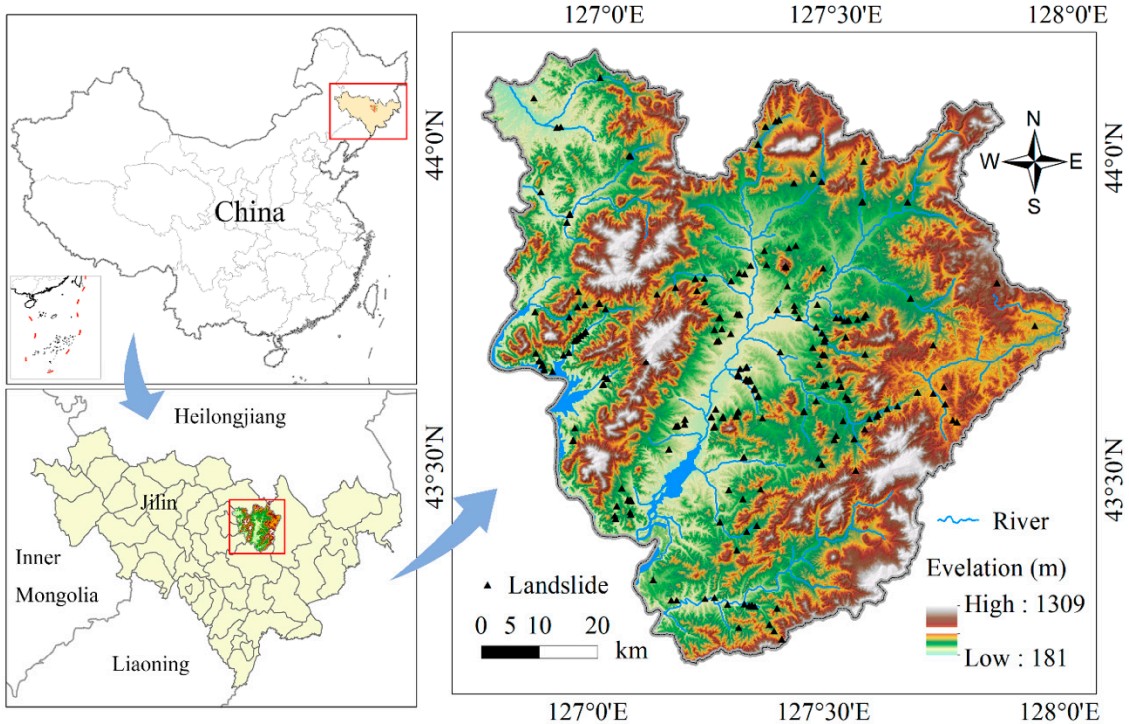

**Figure 1.** Location of the study area.

The general topographic features of the area are hills between 181 m and 1309 m above sea level and their inclination varies from 0° to 57°. The terrain of the central basin is relatively low and flat. Generally speaking, the terrain of the basin is high in the northeast and low in the southwest with great fluctuation. The exposed strata are Paleozoic, Mesozoic, and Cenozoic from old to new. Geomorphology can be divided into tectonic denudation geomorphology, erosion accumulation geomorphology, and volcanic lava geomorphology.

According to the collected data and previous studies [48], Jiaohe County is one of the most serious areas with geological disasters in Jilin Province, which poses a direct threat to the life and property of local residents due to the landslides induced by short-term heavy rainfall and excessive human activities. Farmland, roads, and houses in the research area have been seriously damaged (Figure 2).

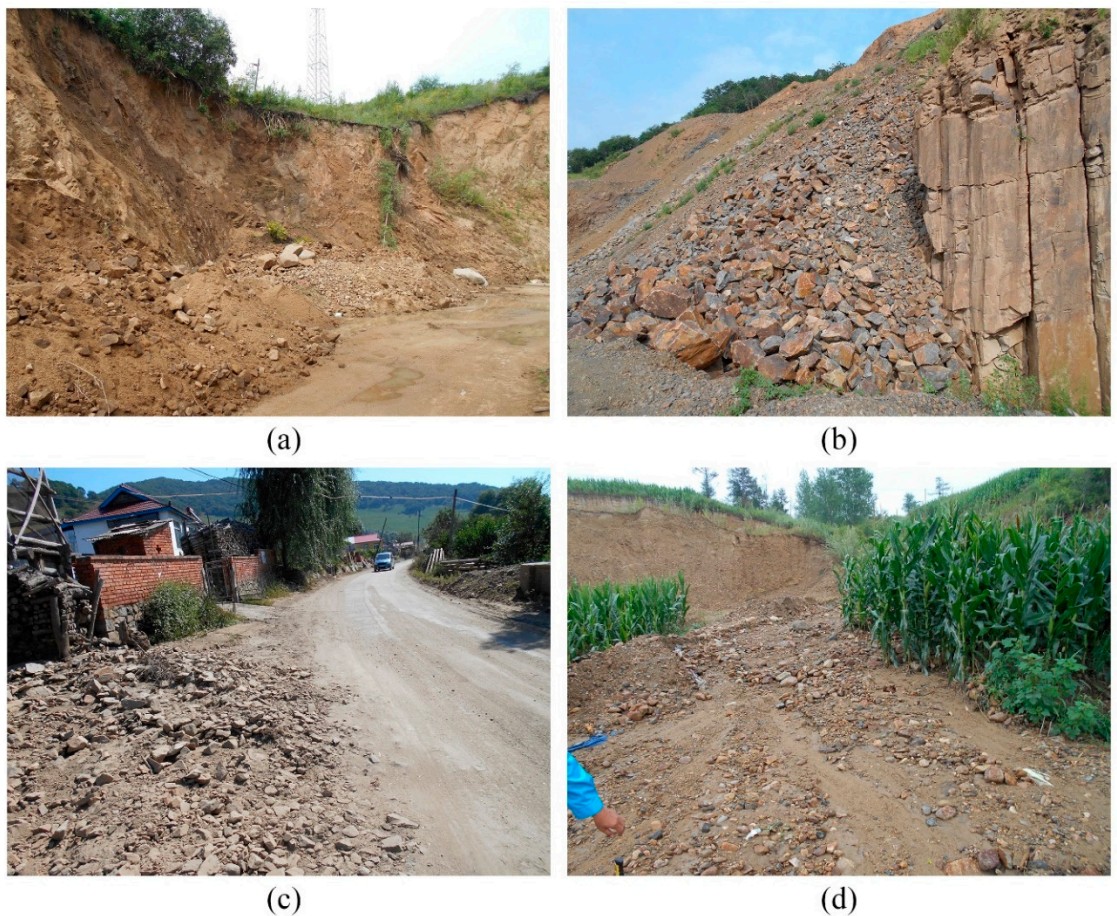

**Figure 2.** Photographs showing the severity of the landslide. (**a,b**) Examples of landslides; (**c**) Deposit and destroyed road; (**d**) destroyed farmland.

## 3. Materials and Methods

The study considered establishing a semi-supervised deep learning framework, where the limited labeled data and abundant unlabeled information in the study area could be devoted to optimizing the LSM process.

Meanwhile, traditional supervised learning methods were applied for comparison and the performance of factors and models were evaluated by a series of indicators. In terms of environment configuration, Tensorflow2.0 was employed in this study to construct and train the landslide susceptibility model. The flowchart used in this study is shown in Figure 3.

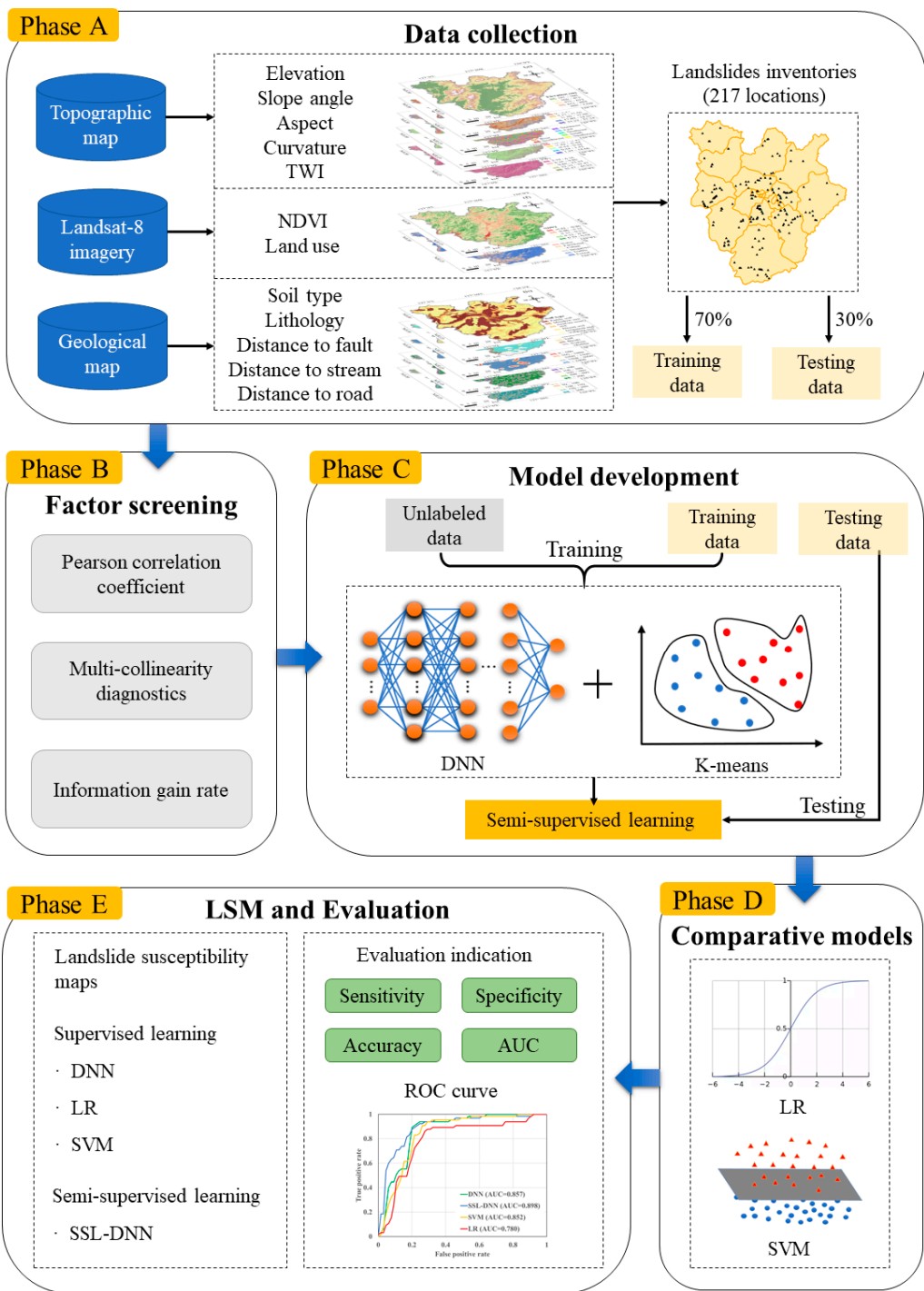

**Figure 3.** The flowchart of this study.

## 3.1. Data Preparation

A map of landslide inventories, as the basis of LSM, affects the regional evaluation results directly [49]. As of 2017, the geological survey has carried out remote sensing interpretation and field investigations on all residential areas, market towns, mines, important public infrastructure, and areas prone to landslides in Jiaohe County. A total of 217 large or small landslides were included in the inventory map. Among them, about 70% of the landslide locations were randomly selected for training, and the remaining 30% for verification.

In the binary classification problem, 0 was used to represent the non-landslide position, and the landslide position was defined as 1. In addition, 3000 unlabeled sites were randomly generated in the research area for the training of semi-supervised model (Figure 4) and could also obtain the corresponding labels in the semi-supervised learning.

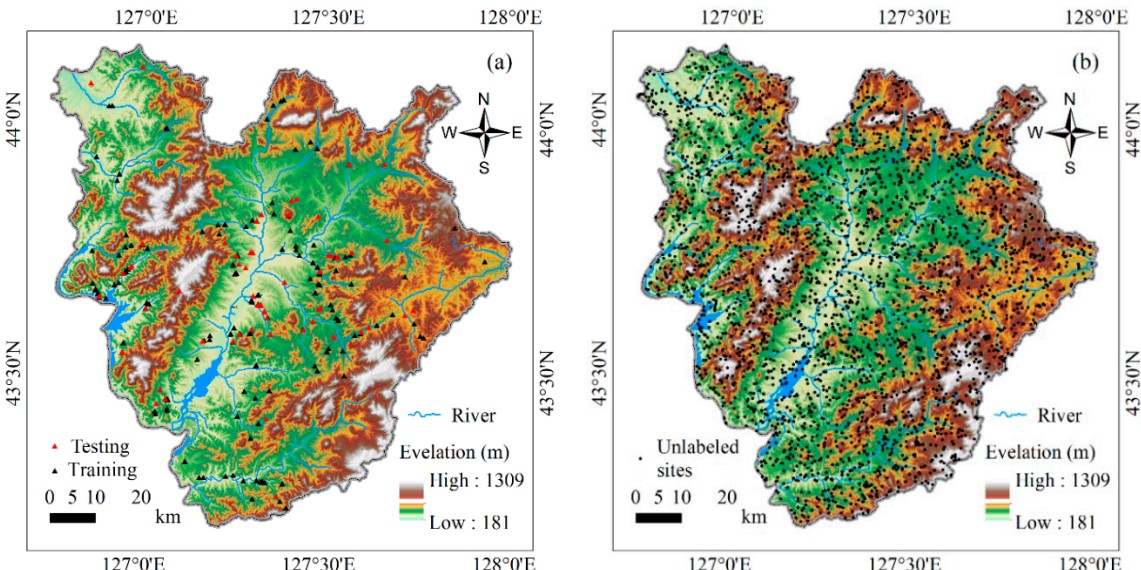

**Figure 4.** Labeled and unlabeled sites in study area. (**a**) Distribution of labeled data; (**b**) Distribution of unlabeled data.

### 3.2. Impact Factors

Identification of factors is a key step in the evaluation of landslide susceptibility [50]. Based on previous studies and relevant data [51–53], 12 impact factors were selected in this paper (Figure 5) including elevation, slope angle, slope aspect, curvature, topographic wetness index (TWI), normalized differential vegetation index (NDVI), land use, soil type, lithology, distance to fault, distance to stream, and distance to road. The data characteristics and sources of the factors are shown in Table 1 and the compilation and production of maps were based on ArcGIS 10.5 software. An almost identical consensus exists in previous studies: There is a nonlinear relationship between susceptibility and the impact factors because of the complexity of the geological environment [54].

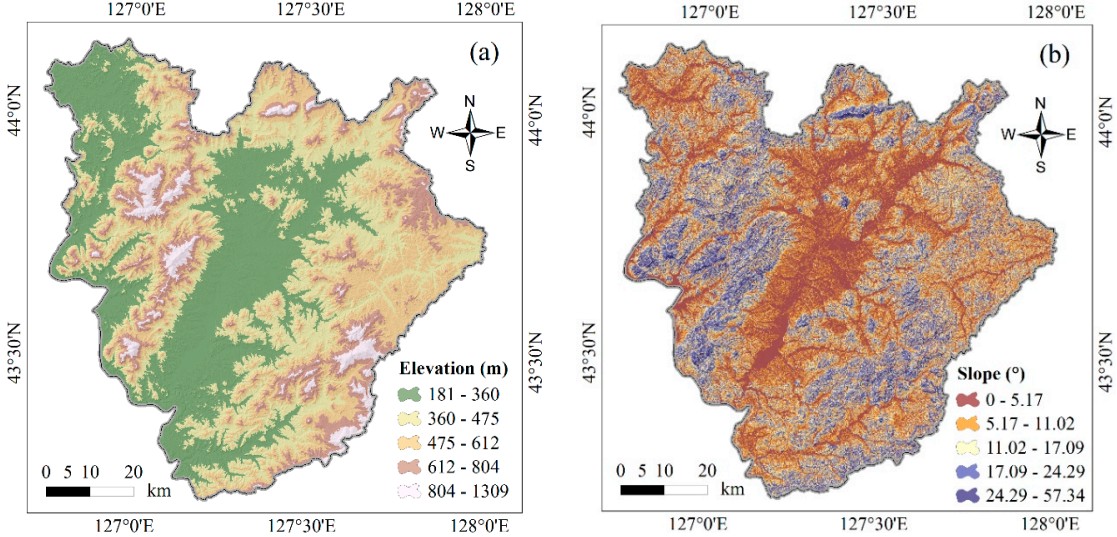

**Figure 5.** *Cont*.

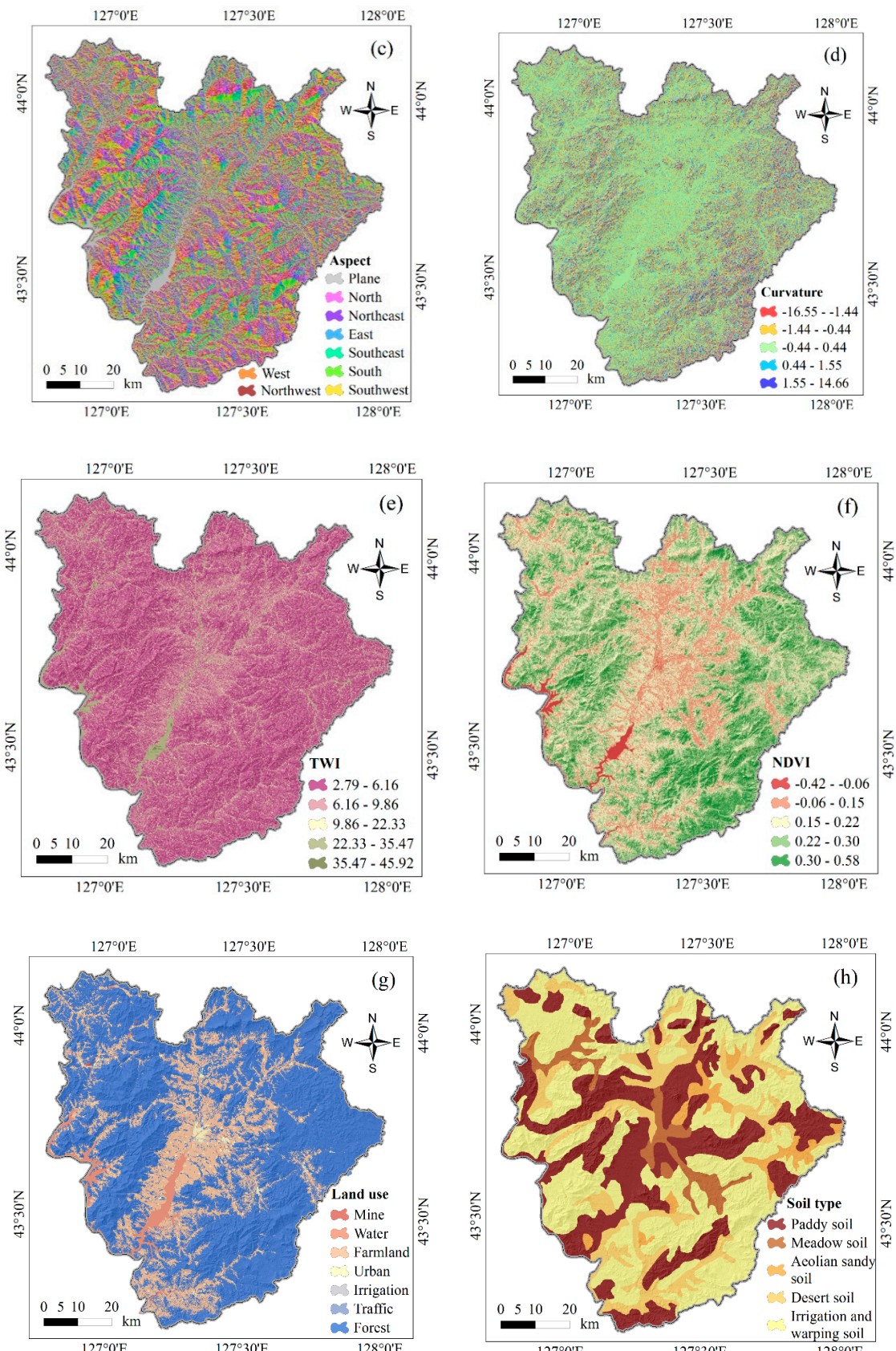

**Figure 5.** *Cont.*

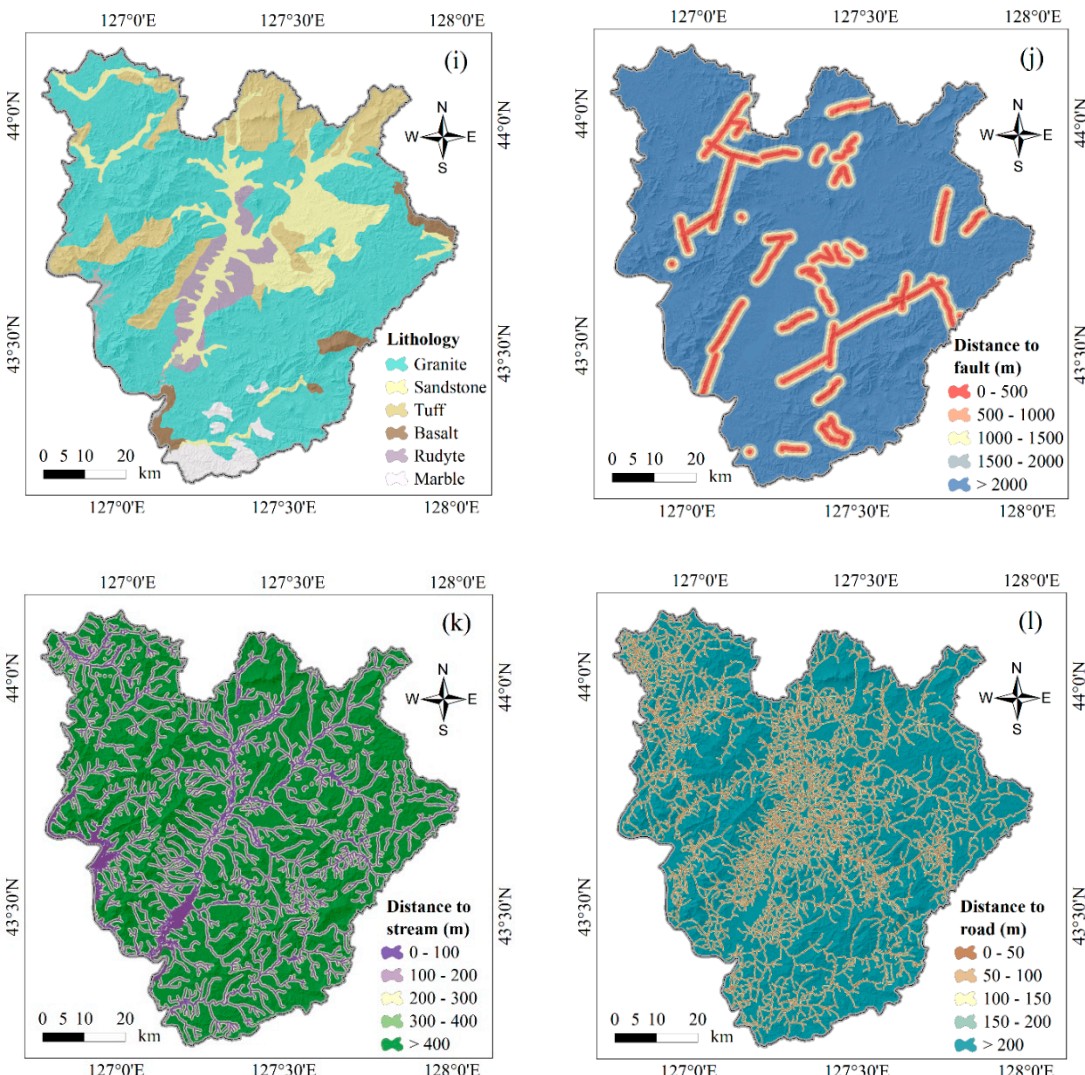

**Figure 5.** Landslide impact factors. (**a**) Elevation; (**b**) Slope; (**c**) Aspect; (**d**) Curvature; (**e**) TWI; (**f**) NDVI; (**g**) Land use; (**h**) Soil type; (**i**) Lithology; (**j**) Distance to fault; (**k**) Distance to stream; and (**l**) Distance to road.

**Table 1.** The conditioning factors used in this study and their significance on landslide occurrence.

| Data Layers | Spatial Resolution | Source | Techniques |
| --- | --- | --- | --- |
| Elevation | $30 \times 30$ m | Topographic map | $30$ m $\times 30$ m DEM |
| Slope angle | $30 \times 30$ m | Topographic map | $\text{Angle} = \arctan\left(\left[\frac{dz}{dx}\right]^2 + \left[\frac{dz}{dy}\right]^2\right) \cdot \frac{180}{\pi}$ <br> The slope angle depends on the rate of change (increment) of the surface in the horizontal $\frac{dz}{dx}$ and vertical $\frac{dz}{dy}$ directions starting from the central pixel. |
| Aspect | $30 \times 30$ m | Topographic map | $\text{Aspect} = \arctan^2\left(\left[\frac{dz}{dy}\right], -\left[\frac{dz}{dx}\right]\right) \cdot \frac{180}{\pi}$ |
| Curvature | $30 \times 30$ m | Topographic map | $30$ m $\times 30$ m DEM |
| TWI | $30 \times 30$ m | Topographic map | $TWI = ln\left(\frac{A_s}{\tan\beta}\right)$ <br> Where $A_s$ refers to the upstream catchment area and $\beta$ represents the slope angle of a certain grid cell. |
| NDVI | $30 \times 30$ m | Landsat-8 | $NDVI = \frac{NIR-IR}{NIR+IR}$ <br> where $NIR$ is near inferred band or band 4 and $IR$ is the infrared band or band 3. |

**Table 1.** *Cont.*

| Data Layers | Spatial Resolution | Source | Techniques |
|---|---|---|---|
| Land use | 1:250,000 | Landsat-8 | Supervised classification (Maximum likelihood) |
| Soil type | 1:250,000 | Geological map | Digitization process |
| Lithology | 1:250,000 | Geological map | Digitization process |
| Distance to fault | 1:250,000 | Geological map | Euclidian distance buffering |
| Distance to stream | 1:250,000 | Geological map | Euclidian distance buffering |
| Distance to road | 1:250,000 | Geological map | Digitization process |

### 3.3. The Framework of Semi-Supervised Deep Learning

### 3.3.1. SSL-DNN for LSM

As a probabilistic problem of dichotomy, the purpose of LSM obtains the probability of landslide hazard at each site according to the known messages [55]. For spatial analysis of geography, there is still valuable information in the study area, and the pixels can be viewed as unlabeled data.

In this study, a semi-supervised deep neural network framework was constructed, and the process of classification and clustering promoted by collaborative learning. The proposed framework is to realize the process that iteratively learns the labeled and unlabeled samples based on clustering and classification algorithm. The goal is to leverage large amounts of unlabeled data to promote performance of the model [56,57]. The GIS platform was devoted to randomly generate 3000 unlabeled samples in the region, along with the initial labeled training and test samples, which would effectively reflect the characteristics of the study area [58].

The workflow can be summarized as the following steps. In step 1, a pre-training DNN is derived from the labeled training samples. In step 2, the unlabeled samples can be predicted to obtain the primary labels using the pre-training DNN. Then, in step 3, the K-means clustering algorithm is selected to cluster the depth characteristics. Meanwhile, the samples that are consistent between the primary and cluster labels are defined as high confidence samples, and the corresponding pseudo-labels are obtained. In step 4, update the labeled and unlabeled samples. In step 5, the pre-training DNN was fine-tuned using the pseudo-labeled and the available labeled samples. Repeat the step until the loss function is less than the preset or the iterations reach a maximum. The flow chart is given in Figure 6.

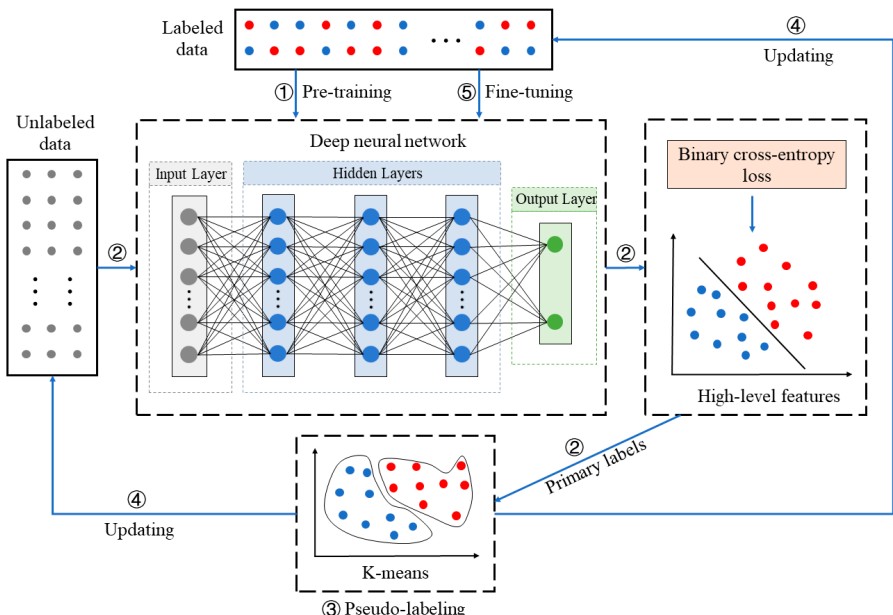

**Figure 6.** Framework of semi-supervised learning.

3.3.2. Pre-Training DNN Architecture

Deep learning, through a non-linear complex model, is a feature learning method that transforms the original data into a higher level and more abstract expression. Deep neural network (DNN) is the basic algorithm of deep learning [59]. The network structure usually consists of an input layer, several hidden layers, and an output layer. DNN discovers complex structures in large datasets by changing the internal parameters, which are used to calculate each layer's representation from the previous layer. So far, various DL architectures (for example, auto-encoders, Restricted Boltzmann Machine (RBM), Convolutional Neural Network (CNN), and Recurrent Neural Network (RNN) have been proposed and perform well in many fields [60,61].

Landslide susceptibility is essentially a nonlinear logistic regression problem. The classification probability in the model can be derived by sigmoid function, that is, the susceptibility of landslide at a certain point. In this study, the DNN model was applied to LSM. Impact factor (IF) became the input signal received in the first layer, and analyzed in the hidden layer. Finally, the prediction category was displayed in the output layer, and two possible labels could be exported: landslide and non-landslide. During the development of the DNN model, the main characteristics set were the number of layers and nodes, which define the depth of the architecture, and the activation and transfer functions. The hidden layers and processing elements were determined by the characteristics of the dataset and the number of training sets.

Based on the above datasets, according to many tests of trial and error, the structure of the DNN (Figure 7), which consisted of a model of three hidden layers including 16 neurons and two output neurons, was established in supervised learning. The network was applied as a pre-training DNN in semi-supervised learning and was also regarded as a supervised learning model for comparison. After the introduction of pseudo labels, the number of neurons was mainly modified to 64.

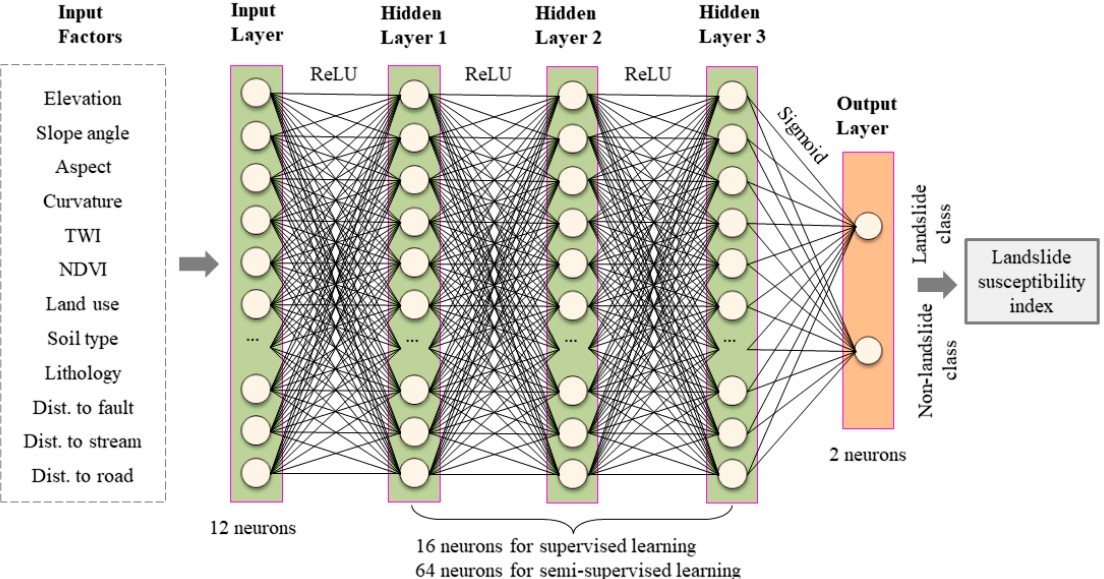

**Figure 7.** Illustration of the DNN for landslide susceptibility.

The activation function introduces nonlinear factors into the neural network, then fits various curves through the activation function. Meanwhile, it converts the input signal of the node into an output signal and stacks it as the input for the next layer. Rectified linear unit (ReLU), as an acclaimed activation function, has been used successfully in a wide range of applications [62,63]. The problem of gradient vanishing can be solved (in a positive interval), and the convergence speed is fast. The analytic expression of the function is:

$$f(x) = \begin{cases} x \ if \ x \\ 0 \ if \ x \end{cases} = \max(x, 0) \tag{1}$$

Sigmoid function is a commonly used nonlinear activation function, and is given by the following equation:

$$f(x) = \frac{1}{1 + e^{-x}} \tag{2}$$

The adaptive moment optimization (Adam) algorithm is employed in the framework as it requires less memory and is efficient in calculation, mainly by combining the advantages of adaptive gradient algorithm (AdaGrad) and root mean square propagation (RMSProp) [64]. As an alternative optimization algorithm, Adam can perfectly replace stochastic gradient descent, and the default parameters are competent to handle most of the problems. First, the gradient and the square of the gradient are calculated for the moving average $m_t$ and $v_t$:

$$m_t = \beta_1 m_{t-1} + (1 - \beta_1) g_t \tag{3}$$

$$v_t = \beta_2 v_{t-1} + (1 - \beta_2) g_t^2 \tag{4}$$

where $\beta_1$ and $\beta_2$ are the exponential decay rate, which default to 0.9 and 0.999, respectively. The exponential moving mean of the gradient $m_0$ and the exponential moving mean of the gradient squared $v_0$ are initialized to 0. Then, the deviation between the two moments are corrected:

$$\hat{m}_t = \frac{m_t}{1 - \beta_1^t} \tag{5}$$

$$\hat{v}_t = \frac{v_t}{1 - \beta_2^t} \tag{6}$$

Finally, update the parameters:

$$\theta_t = \theta_{t-1} - \alpha \cdot \frac{\hat{m}_t}{\sqrt{\hat{v}_t}} + \varepsilon \tag{7}$$

where the default learning rate $\alpha = 0.001$. The parameter $\varepsilon = 10^{-8}$ avoids the divisor approaching zero. The expression indicates that the calculation of the updated step size can carry on the adaptive adjustment from the gradient mean and the gradient square.

The dropout program causes two neurons not to appear in the same dropout network every time, thus preventing the features that are effective only under other specific features, so that the updates of weights no longer rely on the combined action of implicit nodes with fixed relationships [65]. The essence is to discard the units of the neural network, effectively preventing over-fitting. The dropout was set to 0.5 after each hidden layer.

### 3.3.3. Training with Loss Function

As an iterative process, a loss function is necessary in deep learning to measure how good the current forecast of the network (the degree of inconsistency between the predicted value $b$ and the true value $w$) is [66]. It is a non-negative and real-value function, which is usually represented by $L(w, b)$. The smaller the loss function, the better the robustness of the model. Binary cross-entropy loss is the most common loss function in binary classification problems. $y$ and $f(x)$ are the label and probability of sample $x$, respectively, and the equation is defined as:

$$L(w, b) = -\frac{1}{N} \sum_{i=1}^{N} \left( y^{(i)} \log f\left(x^{(i)}\right) + \left(1 - y^{(i)}\right) \log\left(1 - f\left(x^{(i)}\right)\right) \right) \tag{8}$$

### 3.3.4. Pseudo-Labeling via Clustering K-Means

After the initial supervised training of the DNN model, the unlabeled samples can be classified. However, the wrong classification would lead to an error in the training network. Therefore, it is necessary to collect pseudo-label samples with high confidence. Data clustering can automatically divide the same elements into closely related subsets or clusters through quantitative comparison of multiple features. This process is defined as the clustering allocation generated by the clustering algorithm [67].

K-means algorithm is the most popular clustering algorithm [68]. The deep features of unmarked samples were pseudo-labeled in this study. The clustering in LSM is the simplest dichotomy problem (k = 2), namely landslide and non-landslide samples. First, the centroids are selected randomly and calculating the similarity (Euclidean distance $D$) between each sample $x_i$ and the centroids $m_i$. The cluster of the centroids with the highest similarity was determined as the category of the sample. The Euclidean distances between data are defined as:

$$D = \sqrt{\sum_{i=1}^{n}(x_i - m_i)^2} \tag{9}$$

Then, the filtered samples and the corresponding pseudo-labels are introduced into the next iteration. The centroid of each cluster is recalculated, and the high confidence samples are screened again until the clustering meets the expectation.

### 3.4. Comparative Models

### 3.4.1. Support Vector Machine

The SVM model is a classifier based on Vapnik-Chervonenkis (VC) dimension theory and the principle of least structural risk [69]. The learning strategy of SVM is spacing maximization and the optimization algorithm for solving convex quadratic programming. Based on the limited information of the sample, the best compromise between the complexity of the model and the learning ability is sought in order to obtain the best generalization ability.

SVM also includes kernel techniques, which enable it to become a virtual nonlinear classifier. For the nonlinear classification problem in the input space, it can be transformed into a linear classification problem in a dimensional characteristic space by nonlinear transformation.

The kernel function is helpful in transforming the input samples into high-dimensional space so that they can be linearly classified. The kernel function usually has four kinds: linear (LN) polynomial (PL) radial basis function (RBF), and sigmoid (SIG). In this study, RBF was selected because the function has relatively good performance in solving nonlinear regression problems [70]. This is defined as follows:

$$K(x_i, x_j) = \exp(-\gamma \parallel x_i - x_j \parallel^2) \tag{10}$$

### 3.4.2. Logistic Regression

The logistic regression (LR) model is a generalized linear regression analysis model based on statistics [71,72], and is widely applied in binary classification problems. Like linear regression, the goal is to find the correlation coefficient corresponding to each input variable. Then, the logical function (sigmoid) converts any value to a range of 0 to 1. The probability $P(y = 1|x; \theta)$ can be obtained in the LR model as follows:

$$g(\theta^T x) = \theta_0 + \theta_1 x_1 + \theta_2 x_2 + \cdots + \theta_n x_n \tag{11}$$

$$P(y = 1|x; \theta) = g(\theta^T x) = \frac{1}{1 + e^{-\theta^T \cdot x}} \tag{12}$$

where $\theta_0$ is the intercept of the linear regression equation, while $\theta_1, \theta_2, \cdots, \theta_n$ are the coefficients of independent variable $x_1, x_2, \cdots, x_n$.

### 3.5. Feature Metrics

In the framework of landslide evaluation, the impact factors are devoted to represent the region. The noise caused by some factors may lead to the degradation of performance, so it is necessary to check and exclude the impact factors with low predictive ability or repeatability. Therefore, three statistical methods were carried out to screen the characteristics.

The Pearson correlation coefficient is the quotient of the covariance and standard deviation between two variables, which can reflect the degree of linear correlation between two random variables [55]. The value of $r$ is between −1 and 1. When the value is 1 or −1, it represents perfect positive or negative correlation between two random variables, respectively. When the value is 0, it means that the sample correlation coefficient between two random variables is linearly independent, which is denoted as $r$:

$$r = \frac{\sum_{i=1}^{n}(x_i - \overline{x})(y_i - \overline{y})}{\sqrt{\sum_{i=1}^{n}(x_i - \overline{x})^2}\sqrt{\sum_{i=1}^{n}(y_i - \overline{y})^2}} \tag{13}$$

where $n$ is the number of samples; and $x_i$ and $y_i$ are the observed values of point $i$ corresponding to variables $x$ and $y$, respectively, while $\overline{x}$ and $\overline{y}$ are the average values of the samples.

Multi-collinearity diagnosis is an effective tool for determining the linear correlation between two or more variables in a dataset to help select the appropriate characteristic factors [73]. Currently, the method has been applied for a variety of purposes including landslide susceptibility, soil erosion susceptibility, groundwater potential mapping, and so on. The multi-collinearity of the impact factors for landslide susceptibility may lead to the failure of the prediction function. Therefore, multi-collinearity diagnostics was carried out in the impact factor. When tolerance (TOL) is greater than 0.1 or the variance inflation factors (VIF) is less than 10, it indicates that there is no serious collinearity problem in the factor. The formula is as follows:

$$\mathrm{TOL} = 1 - R_j^2 \tag{14}$$

$$\mathrm{VIF} = \frac{1}{TOL} \tag{15}$$

where is explained the determination coefficient of variable $j$ for auxiliary regression model. In this study, the multi-collinearity of 12 impact factors were examined.

The information gain ratio (IGR) evaluates and ranks the importance of input variables due to the wide range of applications in selecting impact factors [74], which can be expressed as follows:

$$E(Y) = -\sum_{i=1}^{n} P(Y_i)log_2(P(Y_i)) \tag{16}$$

$$E(Y|X_i) = -\sum_{i=1}^{n} P(Y_i) \sum_{i=1}^{n} P(Y_i|X_i)log_2(P(Y_i|X_i)) \tag{17}$$

$$IG(Y, X_i) = E(Y) - E(Y|X_i) \tag{18}$$

$$IGR(Y, X_i) = \frac{IG(Y, X_i)}{E(Y)} \tag{19}$$

where $E$ is defined as the entropy value of the impact factor $X_i$ (with $n$ classes) corresponding to the output type $Y$ (landslide and non-landslide). $P(Y_i)$ and $P(Y_i|X_i)$ represents the prior probability of $Y$ and the posterior probability of $Y$ corresponding to $X_i$, respectively.

*3.6. Model Metrics*

The detection of predictive ability is the vital link of LSM to pick out the model with the best performance [75]. Based on previous research, a large number of statistical indicators have been applied to evaluate the machine learning model [28]. In this study, the predictive ability of LSM was carried out on the quantitative. Through the comparison of actual tag and forecast, the calculation of true positives (TP), false positives (FP), false negatives (FN), and true negatives (TN). Then, based on the computed results of the aforesaid four indicators, the sensitivity or specificity, positive predictive value (PPV) or negative predictive value (NPV), accuracy (ACC) and Kappa index can be calculated as follows:

$$\text{Sensitivity} = \frac{\text{TN}}{\text{TN} + \text{FP}}; \text{Specificity} = \frac{\text{TN}}{\text{TP} + \text{FN}} \tag{20}$$

$$\text{PPV} = \frac{\text{TP}}{\text{TP} + \text{FP}}; \text{NPV} = \frac{\text{TN}}{\text{TN} + \text{FN}} \tag{21}$$

$$\text{ACC} = \frac{\text{TP} + \text{TN}}{\text{TP} + \text{TN} + \text{FP} + \text{FN}} \times 100\% \tag{22}$$

In addition, the receiver operating characteristic (ROC) is a curve plotted with the 100-specificity (false positive rate) and sensitivity (true positive rate) on the x-axis and y-axis to draw under the curve (AUC), which is the area under the ROC curve and represents the performance of the two classification models [76].

In order to estimate the significance of the difference among LSMs, the Wilcoxon signed-rank test is more effective than the traditional test with positive and negative signs and supports the pairwise comparison between sensitivity models [77]. When the $p$ value is below the critical threshold (0.05) and the $|z|$ value surpasses 1.96, the null hypothesis will be rejected.

## 4. Results

*4.1. The Analysis of Impact Factors*

The output result of Pearson correlation (Figure 8) is shown. When the threshold of correlation is lower than 0.7, it is considered that there is no obvious positive correlation between parameters, among which the highest correlation between elevation and NDVI was 0.466, indicating that the above parameters are effective for landslide susceptibility.

According to the results in Table 2, there was no significant collinearity between the parameters because the TOL and VIF of all factors were within the threshold. Soil type had a maximum TOL of 0.955 and the VIF of elevation was 1.661. On the other hand, the IGR of the impact factors ranged from 0.016 to 0.301, and NDVI showed the highest value, followed by variables such as distance to road, land use, elevation, etc., while TWI, aspect, slope angle, and other variables had less IGR, but were all greater than 0.01. Therefore, the impact factors proposed in this study were considered as suitable for subsequent modeling.

**Table 2.** Multi-collinearity and IGR among impact factors.

| NO. | Impact Factors | Collinearity Statistics | | IGR |
|---|---|---|---|---|
| | | TOL | VIF | |
| 1 | Elevation | 0.602 | 1.661 | 0.146 |
| 2 | Slope angle | 0.612 | 1.634 | 0.020 |
| 3 | Aspect | 0.930 | 1.076 | 0.019 |
| 4 | Curvature | 0.881 | 1.135 | 0.060 |
| 5 | TWI | 0.729 | 1.372 | 0.016 |
| 6 | NDVI | 0.578 | 1.730 | 0.301 |
| 7 | Land use | 0.615 | 1.625 | 0.205 |
| 8 | Soil type | 0.956 | 1.046 | 0.026 |
| 9 | Lithology | 0.948 | 1.055 | 0.084 |
| 10 | Distance to fault | 0.930 | 1.075 | 0.028 |
| 11 | Distance to stream | 0.747 | 1.338 | 0.086 |
| 12 | Distance to road | 0.650 | 1.539 | 0.263 |

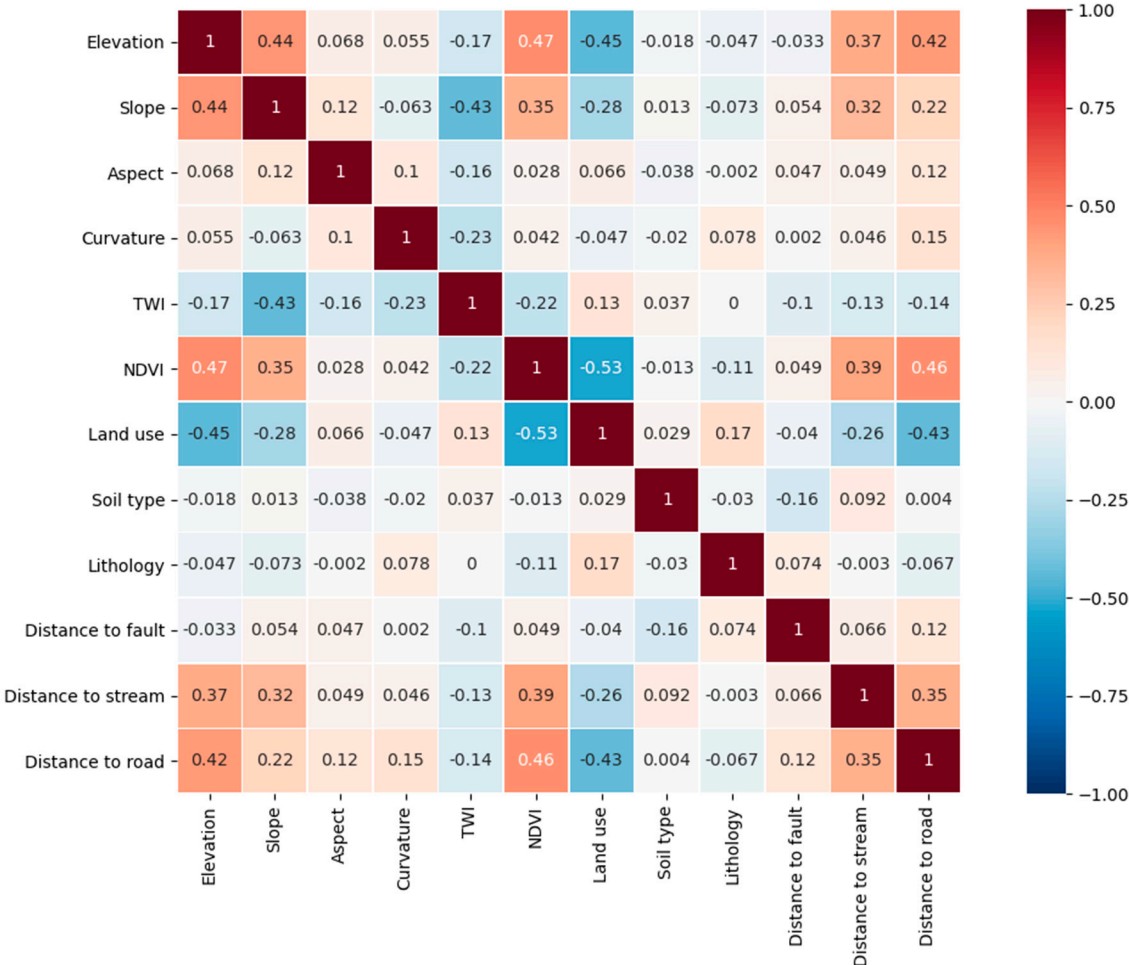

**Figure 8.** The output results of the Pearson correlation matrix.

*4.2. Landslides Susceptibility Assessment*

In this article, PyCharm was selected as a compiler to code the deep learning algorithm through Python. After previous experience and trial-and-error [29,78], the parameters of DNN and SSL-DNN models were determined, as shown in Table 3. Moreover, Figure 9 reflects the variation of accuracy and loss as the iteration progresses. Note that after the number of epochs reached 200, the difference between the training set and the test set in accuracy was less than 0.02, and the loss also tended to be stable, so the networks can be determined to avoid over-fitting.

**Table 3.** Parameter settings of DNN and SSL-DNN.

| Parameters \ Method | DNN | SSL-DNN |
|---|---|---|
| Epochs | 300 | 300 |
| Dropout | 0.5 | 0.5 |
| Learning rate | 0.001 | 0.001 |
| Number of hidden layers | 3 | 3 |
| Dense connection | 16 | 64 |
| Activation function | ReLU | ReLU |
| Optimizer | Adam | Adam |
| Loss function | Binary cross-entropy | Binary cross-entropy |

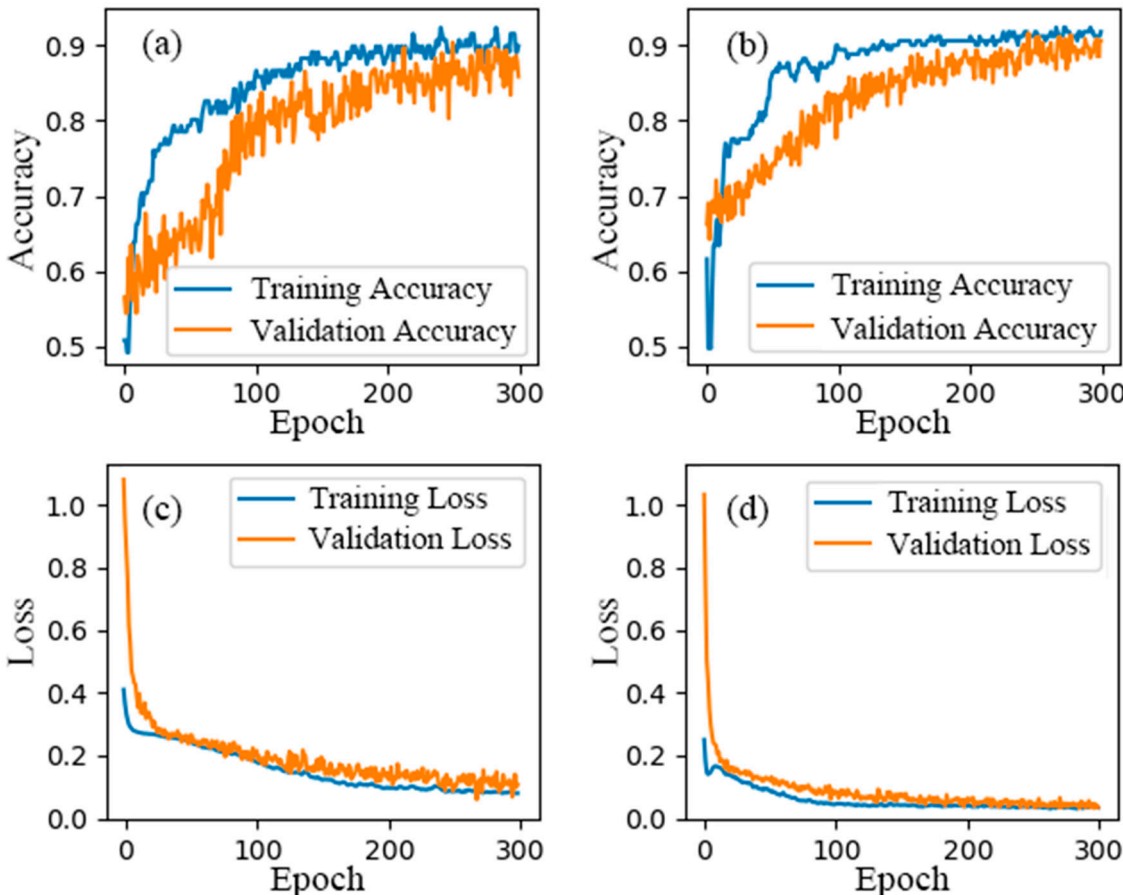

**Figure 9.** The convergence curves. (**a**) Accuracy curves of DNN; (**b**) Accuracy curves of SSL-DNN;
(**c**) Loss curves of DNN; (**d**) Loss curves of SSL-DNN.

After establishing the above semi-supervised and the comparative models, the landslide susceptibility of all pixels in the study area were predicted. These probability raster maps were crucial to visualize the overall quality of LSM Then, the susceptibility index was divided into five grades by the natural fracture method (Figure 10). The higher the hazard grade, the more concentrated the debris flow distribution and the greater the probability of debris flow disaster. The maps consistently indicate that the southcentral and northwestern parts of the study area are hilly and farmland, with landslides most likely to occur. However, for primeval forest with low human activity, the susceptibility of landslide is very low.

Figure 11 shows the relative distribution of the landslide susceptibility classes and the landslide density. The results of susceptibility level showed a similar trend (Figure 11a). In the map produced using the DNN model, the proportion of very high susceptibility was the lowest, accounting for 8.61%, and the other 68.26, 4.40%, 8.12%, and 10.61% were very low, low, moderate, and high probability, respectively. The proportion of each susceptibility level in the LSM predicted by the SSL-DNN model as very low (62.62%), low (4.43%), moderate (4.03%), high (13.55%), and very high (15.37%). Regarding the two shallow learning models, of the results the SVM model generated, about 61.32%, 10.19%, 6.84%, 6.59%, and 15.06% were very low, low, moderate, high, and very high levels, respectively. Finally, based on the LR model, approximately 58.29% of the land area was in the very low susceptibility zone and the other 10.15%, 6.75%, 6.96%, and 17.89% were low, moderate, high and very high probability, respectively. At the same time, the landslide density of the above models was positively correlated with the type of susceptibility, as shown in Figure 11b. In the very high susceptibility zone, the SSL-DNN model had the highest landslide density of 22.21 landslides per 100 $km^2$, while the LR model had the lowest value of 11.15 per 100 $km^2$.

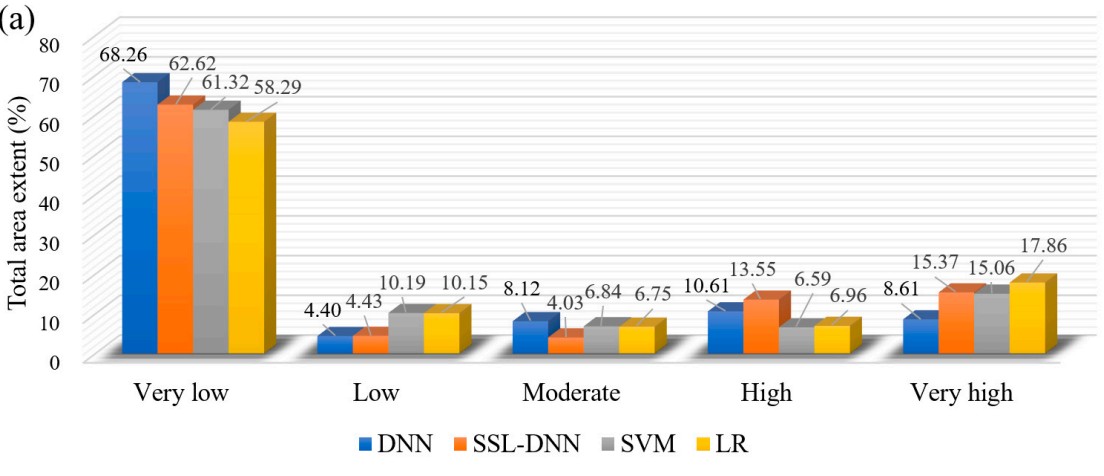

**Figure 10.** Landslide susceptibility maps. (**a**) DNN, (**b**) SSL-DNN, (**c**) SVM, (**d**) LR.

(a)

**Figure 11.** *Cont*.

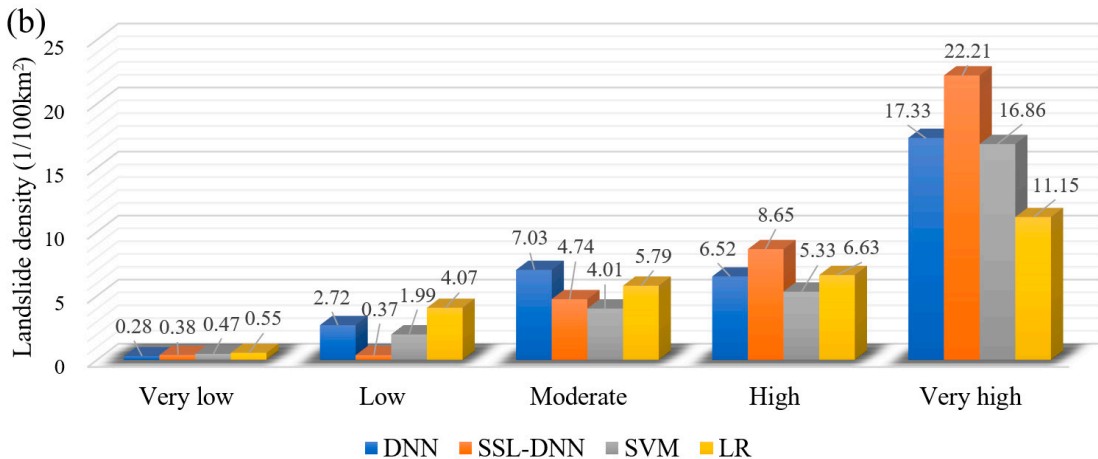

**Figure 11.** Comparison of the distribution using DNN, SSL-DNN, SVM, and LR models. (**a**) Total area covered by each susceptibility map; (**b**) Landslide density of each susceptibility class.

### 4.3. Model Comparison and Validation

In order to evaluate the effectiveness of the above model for LSM, a series of indicators and the testing set data were devoted in the article. The results (Table 4 and Figure 12) indicate that all models produced great fitting and prediction performance.

**Table 4.** Performances of the supervised and semi-supervised learning models.

| Models | TP | TN | FP | FN | PPV | NPV | Sensitivity | Specificity | ACC | AUC |
|---|---|---|---|---|---|---|---|---|---|---|
| DNN | 52 | 54 | 13 | 11 | 0.800 | 0.831 | 0.806 | 0.857 | 0.815 | 0.857 |
| SSL-DNN | 57 | 54 | 8 | 11 | 0.877 | 0.831 | 0.871 | 0.794 | 0.854 | 0.898 |
| SVM | 54 | 52 | 13 | 11 | 0.831 | 0.800 | 0.825 | 0.776 | 0.815 | 0.852 |
| LR | 50 | 48 | 15 | 17 | 0.769 | 0.738 | 0.762 | 0.716 | 0.754 | 0.780 |

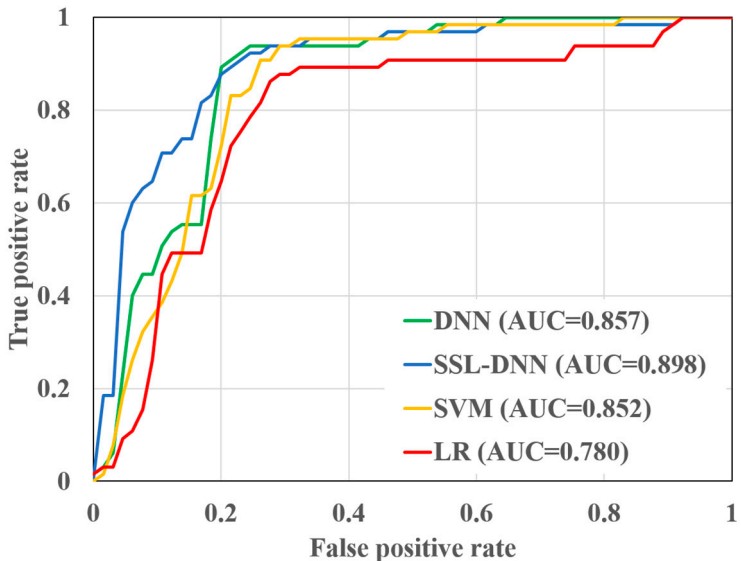

**Figure 12.** ROC curves of the predictive rate.

The PPV (0.877), NPV (0.831), and sensitivity (0.871) values of the SSL-DNN model were all the highest, indicating that the model had the best ability to distinguish landslide pixels from other pixels in the region. The DNN model with a value of specificity (0.857) was superior to other models. As far as the ACC index is concerned, it indicates the accuracy of the whole model, and the SSL-DNN model

(0.846) was considered to be the most accurate model. The ROC curve and AUC value reflect the predictive power of the LSMs. The SSL-DNN model had the best predictive performance (0.898), and the distinction between the DNN and SVM models (0.857 and 0.852) was not large, but were significantly better than the LR model (0.780). Overall, semi-supervised learning achieved better results than supervised learning.

In addition, the difference in the performance results between the LSM models should be statistically evaluated. Wilcoxon signed-rank test results are given based on the paired model (Table 5), and the requirements of $p < 0.05$ and $|z| > 1.96$ were met. This indicates that there are significant differences between each pair of models, which are statistically significant.

**Table 5.** Models pairwise comparison using the Wilcoxon signed-rank test.

| Pairwise Model | *p*-Value | z-Value |
|---|---|---|
| SSL-DNN vs. DNN | 0.000 | −5.925 |
| SSL-DNN vs. SVM | 0.000 | −4.752 |
| SSL-DNN vs. LR | 0.000 | −4.888 |
| DNN vs. SVM | 0.000 | −8.146 |
| DNN vs. LR | 0.000 | −9.097 |
| SVM vs.LR | 0.001 | −4.616 |

## 5. Discussion

### 5.1. Impact Factor on Control of Landslide

The study on the factors for the geological conditions is always the focus of landslide susceptibility evaluation. According to the spatial heterogeneity, the effect of controlling landslide is different due to factors such as the geology, soil, topography, climate, and land use. Highly similar factors can be screened out to avoid influence on the classification process by using correlation and collinearity and the IGR can evaluate the rank of input factors and understand the tendency of susceptibility.

In this study, the NDVI was considered to be the most critical factor in determining landslide susceptibility. Landslides mostly occurred on exposed slopes, and dense woodlands were almost free from disasters. Land use and cover indicated that landslides occurred mostly on sloping farmland and around roads. On one hand, this implies that cultivated land and engineering activities have a certain catalytic effect on the occurrence of landslide. On the other hand, farmland and roads were the main threats of disaster, which was also consistent with the previous survey.

### 5.2. Susceptibility in SSL-DNN

The experimental results of the above models were compared, and the validating outcome (Table 4) indicated that the results of the DNN (0.857) and SVM (0.852) models in supervised learning were obviously better than those of the LR model (0.780). Deep learning, as an artificial neural network with multiple hidden layers for nonlinear transformation of the original data was developed to abstract implicit features. There is no doubt that the application of DNN in LSM was successful. However, from the results of supervised learning, the performance of the DNN model did not seem to reveal much advantage over the SVM model. According to previous studies, the SVM model can maintain the capacities of prediction and generalization when the sample size is insufficient. Furthermore, the DNN model tends to overfit the training data relatively. Even a simple linear model may be superior to deep network models. Predecessors have suggested that efforts should be devoted to find ways to address the problem of insufficient datasets [43,79].

In this study, the dropout layer can alleviate the above problem to some extent. Furthermore, 3000 unlabeled points were randomly selected to obtain corresponding pseudo-tags for semi-supervised learning. Comparing the results of semi-supervised learning with those of supervised learning, the semi-supervised deep learning achieved the optimal performance (0.898). The purpose of pseudo

labels in LSM is to obtain the binary classification results of unlabeled points. This avoids the problem that too many predictive categories will introduce large error signals. It seems to be inferred that the regression problem of dichotomies is more suitable for semi-supervised learning applications. For the selection of unlabeled data, while ensuring that the unlabeled information fully reflected the regional characteristics, the extra cost of calculation caused by too much data was avoided as much as possible.

In terms of the degree to which the models were optimized, there were significant differences between the results of each model in Table 5. This indicates that the benefit brought about by unlabeled samples and semi-supervised learning was also valuable. The unlabeled data produced an effect like regularization, which reduced the over-fitting of the network under the limited labeled data, and enhanced the ability of generalization. Furthermore, the fitting ability of the classifier would determine the upper limit of semi-supervised learning to some extent. The DNN model was selected as the basic model for semi-supervised training, and the expansion of the sample enabled the network to more fully capture the complexity of landslide distribution. Intuitively, the SSL-DNN model organically integrated the distribution information hidden in the unlabeled sample data. Meanwhile, the multi-layer nonlinear structure enabled it to have strong capability of feature expression and modeling for complex tasks.

Although the SSL-DNN model can be confirmed as an alternative with an ideal degree of predictive accuracy, there are still several limitations in the experiment. Due to the lack of theoretical basis, the selection of the super-parameters and network design were also a considerable challenge. Many trial and error tests may be required for a static structure. Second, to facilitate the classification of pseudo labels, the cluster number of K-means algorithm was fixed at 2, which may not obtain the best clustering result. In addition, we did not discuss the impact of the difference in the number of unlabeled samples, and the comparison process of univariate changes was difficult to achieve due to the constant fine-tuning of iterations and network structure. According to the previous deep learning research [80,81], 3000 extended points are enough to meet the complexity of LSM.

In general, although the effect of the model was also limited by the accuracy of layers and sampling process, the potential of SSL-DNN in the geospatial analysis of susceptibility cannot be ignored. In future work, the optimization of structural parameters and the automation of procedure would be the main areas of deep learning and semi-supervised learning for LSM.

## 6. Conclusions

In this paper, we focused on the problem of limited labeled samples to develop a semi-supervised deep learning. A framework (SSL-DNN) combining the DNN and K-means algorithm was proposed for LSM. In addition, a geospatial database was also established for Jiaohe County, Jilin Province, China including historical records of landslides and 12 related variables. Among them, the NDVI put up the highest predictive capacity, followed by land use and distance to road.

Experimental outcomes showed that the SSL-DNN model had the best performance (0.898), and the method was obviously superior to supervised learning methods such as DNN (0.857), SVM (0.852), and LR (0.780) models. Therefore, it is beneficial to introduce pseudo-label samples for training a DNN. Semi-supervised clustering incorporates a large amount of unlabeled information into the modeling and fully explores the potential of deep learning in spatial landslide modeling.

In conclusion, this work could assist in land-use planning and in the development of effective strategies for landslide disaster mitigation and prevention. Moreover, the semi-supervised deep learning model could be applied to the spatial prediction of other natural disasters (such as flash floods, forest fires, and gully erosion) and provide a feasible direction in the under-sample area.

**Author Contributions:** Conceptualization, S.Q. (Shengwu Qin) and J.Y.; Methodology, J.Y.; Software, J.Y.; Validation, W.C., Y.C., and G.S.; Formal analysis, J.Y.; Investigation, S.Q. (Shuangshuang Qiao) and Q.M.; Data curation, J.Y.; Writing—original draft preparation, J.Y.; Writing—review and editing, J.Y., W.C., and S.Q. (Shuangshuang Qiao); Visualization, J.Y.; Supervision, S.Q. (Shengwu Qin); Project administration, S.Q. (Shengwu Qin); Funding acquisition, S.Q. (Shengwu Qin). All authors have read and agreed to the published version of the manuscript.

**Funding:** This research was funded by the Jilin Provincial Science and Technology Department (no. 20190303103SF) and the National Natural Science Foundation of China (grant no. 41977221).

**Conflicts of Interest:** The authors declare no conflict of interest.

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
