# Peer review of "Assessment of Landslide Susceptibility Combining Deep Learning with Semi-Supervised Learning in Jiaohe County, Jilin Province, China"

_applsci, doi:10.3390/app10165640_

Round 1

Reviewer 1 Report

I think the authors carried out a large amount of work at modeling for the study area. The paper is generally well-written in an understandable way, and the use of English is good. But the methodology of the framework based on deep learning neural network is incomplete with substantially missing information (e.g., design, optimization of the model). I have included several suggestions and recommend the manuscript for publication after the following moderate changes:

  1. Line 20-22: can you reformulate using quantitative results? what do you mean by promising I line 23? You should provide statistical results for justification.
  2. There are many well established traditional pixel-based models (e.g., CART(Choubin, al. 2018.)QRF (Bhuiyan et al. 2020), BART (Yang et al. 2020), Boosted tree etc)  which successfully applied in remote sensing , hazard mitigation, wildfire protection etc. Why did you use DL neural network over other AI-based methods? In the introduction section, You can explain the merit of other machine learning techniques(more literature review requires) in hazard application, which will help to reader why you chose your proposed techniques is chosen as the baseline product for this study? In your paper, I believe You can take the merits of their output.

Choubin, al. 2018. Precipitation forecasting using classification and regression trees (CART) model: A comparative study of different approaches. Environ. Earth Sci. 2018, 77, 314.

Yang et al. 2020, E.N. Quantifying uncertainty in machine learning-based power outage prediction model training: A tool for sustainable storm restoration. Sustainability 2020, 12, 1525.

Bhuiyan, et al. 2020,. Advanced wind speed prediction using convective weather variables through machine learning application. Appl. Comput. Geosci. 2019, 1, 100002.

  1. In 3. Materials and Methods section, Figure 3: I do appreciate the scheme which is helpful. In addition, Line 121-127 you need citations which are completely missing. Can you explain about this statement “Recently, DNN has achieved excellent results in LSM? How did you label your site? Are the data quality controlled? When do they measure?
  2. What are the criteria for selecting predictors like (NDVI), land use, soil type? What is the relative importance of these predictors in improving the performance of machine learning techniques? Are they only selected based on previous works? You should mention previous work in detail and provide proper citations. Why you did not use soil moisture and temperature?
  3. In methodology section, can you explain about your pre-training DNN model? Authors need to provide significant information such as epoch, learning rate, learning momentum, weight decay. You ca provide a table for the The tuned hyperparameters for DNN model.  How did you create robust the model without showing any fundamental results? Several plots are mandatory to  justify the model set up

epoch vs loss

Epoch vs accuracy

  1. Deep learning neural network characterized by more than single hidden layer. How do you chose layer number? Is Layer densely or parsley connected which is also missing in your manuscript?
  2. Could you please produce few feature map and includes in you paper which will create a proper idea how you initialize your DL model
  3. There is also no information on avoiding overfitting. One of the challenges in data driven methods is overfitting (i.e. the method is so fine tuned to the training data, and has larger errors when applied to new datasets). I don’t see any discussion of this in the paper. Are there noticeable differences between the performance of the method during training and validation?

Author Response

Dear Reviewer:

Thanks for your letter and comments concerning our manuscript “Assessment of landslide susceptibility combining deep learning with semi-supervised learning in Jiaohe County, Jilin Province, China” (applsci-892690). Great appreciation goes to the editorial board and reviewer who kindly give excellent suggestions on writing and technical issues. We have revised this paper according to the reviewer’ suggestions and tried our best to perfect it. The revised manuscript has used the "Track Changes" function to annotate the modifications in the article. And we highlighted the changed parts in yellow in revised paper. The main corrections in the paper and the responds to your comments are as following:

Reviewer 2 Report

Almost every sentence needs correction in English grammar, punctuation, subject-verb agreement, and/or correct word usage and phrases. 

Your comparison of models was interesting, but mathematically complex and sophisticated.

Your work will mainly be interesting to colleagues in your narrow area of expertise, and not easily understood or adaptable to policy making. If you have a goal of influencing decision making and policy related to lowering the risks of landslides and saving lives and reducing property damage, you will need to develop models that are less formal and more parsimonious. 

Overall, your presentation, especially your visuals, was outstanding. 

Author Response

(The authors gave the same response as above.)

Round 2

Reviewer 1 Report

Authors significantly improved the manuscript by addressing all the comments.